# SEAL: SEmantic-Augmented Imitation Learning via Language Model

## Abstract

Hierarchical Imitation Learning (HIL) is a promising approach for tackling long-horizon decision-making tasks. While it is a challenging task due to the lack of detailed supervisory labels for sub-goal learning, and reliance on hundreds to thousands of expert demonstrations. In this work, we introduce SEAL, a novel framework that leverages Large Language Models (LLMs)'s powerful semantic and world knowledge for both specifying sub-goal space and pre-labeling states to semantically meaningful sub-goal representations without prior knowledge of task hierarchies. SEAL employs a dual-encoder structure, combining supervised LLM-guided sub-goal learning with unsupervised Vector Quantization (VQ) for more robust sub-goal representations. Additionally, SEAL incorporates a transition-augmented low-level planner for improved adaptation to sub-goal transitions. Our experiments demonstrate that SEAL outperforms state-of-the-art HIL methods and LLM-based planning approaches, particularly in settings with small expert datasets and complex long-horizon tasks.

## 1 Introduction

The advancement of LLMs brings transformative change to how agents learn to interact and make decisions (Brohan et al., 2023; Wang et al., 2023). LLMs like GPT-4 (Achiam et al., 2023) possess remarkable semantic understanding ability (Liu et al., 2023), human-like reasoning capability (Wei et al., 2022), and rich common sense knowledge (Bubeck et al., 2023), enabling them extracting insights from language instructions to support decision-making agents (Eigner & Händler, 2024).

A promising path towards LLM-assisted decision-making is to improve Deep Reinforcement Learning (DRL) agents through reward design (Kwon et al., 2023; Ma et al., 2023). Yet DRL often suffers from sample inefficiency and still requires extensive interactions with the environment particularly for tasks with long planning horizons and sparse rewards, where agents are rewarded only upon task completion (Zhang et al.). Alternatively, Imitation Learning (IL) can be trained to learn generalizable policies from expert demonstrations (Schaal, 1996). By learning from successful state-action pairs, IL avoids the sample-expensive exploration and exploitation required in DRL. While IL performance can still be limited in long-horizon tasks due to compounding errors with accumulated errors leading to significant deviations from desired trajectories (Nair & Finn, 2019). *Hierarchical Imitation Learning* (HIL) (Le et al., 2018a) leverages the sub-goal decomposition of long-horizon tasks into a multi-level hierarchy and reduces the relevant state-action space for each sub-goal such as goal-states (Ding et al., 2019) and Task IDs (Kalashnikov et al., 2021). To address this, language instruction has come as an aid for sub-goal specification, as such instruction can be both informative and flexible (Stepputtis et al., 2020). Language-conditioned HIL approaches either train the high-level sub-goal encoder and low-level policy agent separately (Prakash et al., 2021) or jointly (Hejna et al., 2023). Though both methods have achieved impressive results, learning language-based sub-goals remains challenging, as it requires a large number of expensive sub-goal labels (Chevalier-Boisvert et al., 2018a). To overcome this, various methods have been proposed to infer both sub-goal boundaries and supervision-free representations (Garg et al., 2022; Jiang et al., 2022; Kipf et al., 2019; Simeonov et al., 2021). Despite these efforts, these language instructions are unstructured, and often fall short of generalization to newer tasks while not seamlessly integrated into IL policy training (Wang et al., 2019a; Mees et al., 2022).

The capabilities brought by Large Language Models (LLMs) spark new promises for tackling such instruction challenges in IL and especially HIL settings. It has shown leveraging LLM's strong reasoning and semantic abilities help break down complex, ambiguous language instructions into manageable steps for a high-level plan (Huang et al., 2022; Ahn et al., 2022; Huang et al., 2023). LLM excels at emulating human-like task decomposition, (Huang et al., 2022; Wei et al., 2022) This ability has already been harnessed by researchers to produce high-level plans based on textual task instructions (Ahn et al., 2022; Prakash et al., 2023; Huang et al., 2023). However, while these plans reflect some task hierarchy, they are not directly executable due to the reliance on pre-trained low-level policy agents for primitive actions (Prakash et al., 2023). Moreover, most high-level plans are static and require frequent LLM calls to update as states change, which is highly costly (Song et al., 2023). These challenges limit applicability of LLM-based approaches in assisting HIL. Motivated by these promises and challenges, we want to address the following question: *"Can pre-trained LLMs serves as a prior to define task's hierarchical structure, establish sub-goal library autonomously, and use them to closely guide both high-level sub-goal learning and low-level agent?"*

In this paper, we explore the possibility of using LLM-generated high-level plans to assist both sub-goal learning and policy training in Hierarchical Imitation Learning. We introduce *SEmantic-Augmented Imitation Learning* (**SEAL**), a novel hierarchical imitation learning framework that utilizes pretrained LLMs to generate semantically meaningful sub-goals. Give the fact that sub-goal are also oftenwise labeled by human and understandable by LLMs, SEAL represents each sub-goal with one-hot latent vectors, and employs these representations to convert expert demonstration states into supervisory labels for high-level latent variable learning. To enhance the learning process, SEAL features a dual-encoder structure for sub-goal representation. One encoder learns sub-goal vectors in a supervised manner using LLM-provided labels, while the other leverages unsupervised Vector Quantization (VQ) (Wang et al., 2019b) to map expert demonstration states to latent sub-goal representations. The effectiveness of the learned latent variables is evaluated within the task environment by comparing their contribution to selecting optimal actions, with success rate as the confidence metric. We design a weighted combination of the two encoders' outputs as the final sub-goal representation, thereby reducing over-reliance on a weaker encoder, mitigating over-fitting, and improving overall robustness. Additionally, we present a transition-augmented lower-level policy agent that prioritizes intermediate states corresponding to sub-goal transitions by assigning higher weights, reflecting the hierarchical structure of long-horizon tasks. Extensive experiments on two tasks *KeyDoor* and *Grid-World* and show that SEAL can outperform several state-of-the-art supervised and unsupervised HIL approaches. To summarize, our main contributions include:

- We propose SEAL, a novel hierarchical imitation learning framework that leverages Language Models to generate high-level plans and provide semantically meaningful sub-goal representations without any prior knowledge of task hierarchical structure.

- To enhance SEAL's effectiveness, we introduce a dual-encoder structure that combines supervised LLM-based sub-goal learning with unsupervised VQ-based representations, ensuring reliability and robustness in sub-goal learning. Additionally, SEAL incorporates a transition-augmented low-level planner for better adaptation to challenging intermediate states where sub-goal transitions occur.

- We demonstrate that SEAL outperforms several state-of-the-art HIL approaches, particularly on small expert datasets, and shows superior adaptability to longer-range composition tasks and task variations.

## 2 RELATED WORKS

**Imitation Learning.** Imitation Learning encompasses two primary approaches: Behavioral Cloning (BC) (Bain & Sammut, 1995) and Inverse Reinforcement Learning (IRL) (Ng et al., 2000). BC relies on a pre-collected expert dataset of demonstrations, where the agent learns to mimic the actions in an offline manner. While BC is simple to implement, it is prone to compounding errors, particularly when the agent encounters states not present in the expert's demonstrations (Zhang, 2021). In contrast, IRL methods (Ho & Ermon, 2016; Reddy et al., 2019; Brantley et al., 2019) involve interacting with the environment to collect additional demonstrations, aiming to infer the underlying reward function that the expert is optimizing. The agent then learns by optimizing this inferred reward. However, IRL approaches are more challenging to implement (Kurach et al., 2018), typ-

ically requiring more computational resources and data. In this work, we primarily adopt the BC architecture in a hierarchical setting, while incorporating insights from IRL by using environment interactions to validate the reliability of learned latent sub-goal variables.

**Bi-Level Planning and Execution.** Hierarchical Imitation Learning (HIL) enhances the ability of imitation learning agents to tackle complex, long-horizon tasks by breaking them down into smaller sub-goals and conditioning the agent's behavior on those sub-goals. The high-level agent chooses the sub-goals, while the low-level agent learns to accomplish specific controls under selected sub-goals (Jing et al., 2021). Many HIL approaches, such as Hierarchical Behavior Cloning (Le et al., 2018a) and Thought Cloning (Hu & Clune, 2024), rely on supervisory labels for sub-goal learning, but such annotations are often difficult to obtain. To address this limitation, unsupervised methods like Option-GAIL (Jing et al., 2021), LOVE (Jiang et al., 2022), SDIL (Zhao et al., 2023), and CompILE (Kipf et al., 2019) have been developed to infer sub-goals directly from expert trajectories. However, the lack of labeled guidance in these approaches makes meaningful sub-goal discovery more challenging and hence reduces the reliability of the learned policies.

**LLMs for Planning.** Large Language Models (LLMs) have demonstrated significant potential in decision-making processes. Direct generation of action sequences usually do not lead to accurate plans (Silver et al., 2022; Valmeekam et al., 2023; Kambhampati et al., 2024). Recent studies have successfully utilized LLMs to decompose natural language task instructions into executable high-level plans, represented as a sequence of intermediate sub-goals (Ahn et al., 2022; Prakash et al., 2023; Huang et al., 2023). While LLMs can be also applied to translate user-given language instructions to symbolic goals (Mavrogiannis et al., 2024; Xie et al., 2023) Additionally, LLMs can function as encoders, identifying current sub-goals based on both observations (sometimes images) and language task descriptions to facilitate high-level plan execution (Fu et al., 2024; Malato et al., 2024; Du et al., 2023). However, these approaches typically still depend on pre-trained low-level planners for generating executable primitive actions. In this work, we leverage LLM-generated high-level plans to assist in learning both sub-goals and low-level actions simultaneously.

## 3 PRELIMINARY

In this paper, we look into the long-horizon, compositional decision-making problem as as a discrete-time, finite-step **Markov Decision Process (MDP)**. MDP can be represented by a tuple $(\mathcal{S}, \mathcal{A}, \mathcal{T}, r, \gamma)$, where $\mathcal{S}$, $\mathcal{A}$ denotes the state and action space, $\mathcal{T}(s_{t+1}|s_t, a_t) : \mathcal{S} \times \mathcal{A} \to \mathcal{S}$ denotes the transition function, $r : \mathcal{S} \times \mathcal{A} \to \mathbb{R}$ is the reward function and $\gamma \in [0, 1]$ is the discount factor.

In standard settings of Hierarchical Imitation Learning (HIL), instead of having access to the reward function $r$, the agent has access to a dataset of expert demonstrations $\mathcal{D} = \{\tau_1^e, \tau_2^e, ..., \tau_N^e\}$, which contains $N$ expert trajectory sequences consisting of state-action pairs $\{(s_t, a_t)\}$, where $s_t \in \mathcal{S}$, $a_t \in \mathcal{A}$, $T$ is the time horizon for planning, $0 \leq t \leq T$. In this paper, the expert trajectories are not labeled with any rewards nor subhorizon segments. We assume HIL agents operate in a two-level hierarchy though our method can also be applied to problems with more levels:

- **High-level Sub-goal Encoder** $\pi_H(g_t|s_t)$: Selects a sub-goal $g_t \in \mathcal{G}$ based on the current states $s_t$, where $\mathcal{G}$ is the space of sub-goals.

- **Low-level Policy Agent** $\pi_L(a_t|g_t, s_t)$: Executes actions conditioned on both the current state $s_t$ and sub-goal $g_t$.

In this work, we focus on settings where agents lack access to the sub-goal space $\mathcal{G}$, relying instead on an oracle full task instruction $\mathcal{M}$ in natural language. While well-defined $\mathcal{G}$ aids efficient HIL agent learning (Hauskrecht et al., 2013), its acquisition is difficult due to missing task-specific knowledge (Nachum et al., 2018; Kim et al., 2021). Natural language task instructions, though easier to obtain as they are common-used commands from human (Stepputtis et al., 2020), are hard to map to hierarchical structures due to their complex and ambiguous nature (Zhang & Chai, 2021; Ahuja et al., 2023; Ju et al., 2024). In this work, we investigate leveraging LLMs to parameterize $\mathcal{G}$ from language instructions with its powerful semantic and world knowledge, and pre-label states in $\mathcal{D}$ to guide effective learning of $\pi_H$ and $\pi_L$ in hierarchical imitation learning.

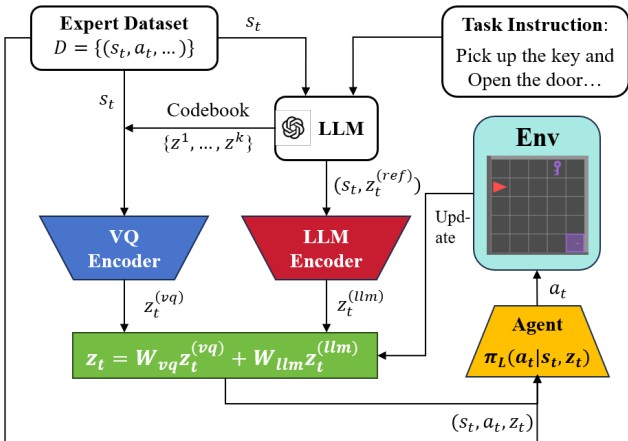

Figure 1: **Overview of SEAL Architecture:** The LLM aids in discovering sub-goal spaces for the task by semantically decomposing the full-task instruction and labeling each state with a reference latent vector that represents its corresponding sub-goal. These reference labels are then used to train a high-level sub-goal encoder, which works in conjunction with an unsupervised VQ encoder.

## 4 SEAL FOR HIERARCHICAL IMITATION LEARNING

The key idea of SEAL is to learn high-level sub-goal representations using supervisory labels generated by LLMs. In previous works, such labels were typically provided by human experts via instructions (Pan et al., 2018; Le et al., 2018a), making them expensive to obtain. However, with the assistance of LLMs, we introduce an efficient and reliable method to automatically generate labels that map states to sub-goals. Specifically, LLMs are used to semantically extract a high-level plan from the full-task language instruction $\mathcal{M}$ and map states in expert demonstrations to sub-goals within this plan. Using these learned sub-goal representations, the model then learns the corresponding low-level actions. An overview of our SEAL framework is illustrated in Fig. 1.

### 4.1 PRETRAINED LLMs FOR GUIDING SUB-GOALS LEARNING

Our key design of leveraging LLMs to guide high-level sub-goals learning can be divided into two stages: (i) Use LLM-generated high-level plan based on full-task instruction as **sub-goal space** (ii) Use LLMs to encode states in expert demonstrations to sub-goal representations.

**Derive Sub-goal Space of Task** Prior works have demonstrated that LLMs can establish a meaningful chain of sub-tasks from task instruction as high-level plan (Huang et al. (2022); Prakash et al. (2023); Singh et al. (2023)). Yet few of them incorporate it with Hierarchical Imitation Learning (HIL). In SEAL, we use LLMs to specify the unknown sub-goal space $\mathcal{G}$ in HIL formulations. Feeding LLMs with the full-task language instruction $\mathcal{M}$, we notice that the decomposed sub-goals in high-level plan naturally consist of a language-based sub-goal set: $\{\hat{g}^1, \hat{g}^2, ..., \hat{g}^K\} = f_{llm}(\mathcal{M})$, where $K$ is the total number of generated sub-goals. We treat this estimated sub-goal dataset as the finite sub-goal space: $\mathcal{G} = \{\hat{g}^1, \hat{g}^2, ..., \hat{g}^K\}$.

**Labeling Sub-goals for States in Expert Dataset** After devising the sub-goal space $\mathcal{G}$ with LLM-generated sub-goals, we use them to map states $s_t \in \mathcal{D}$ to a sub-goal latent space. These LLM-defined labels guide the high-level encoder to learn task-relevant sub-goal representations. To parameterize the language-based sub-goals $\hat{g}^i \in \mathcal{G}$ and facilitate learning, we establish a codebook $\mathcal{C} = \{z^1, z^2, ..., z^K\}$, where each latent variable $z^i \in \mathbb{R}^K$ is a one-hot vector (*i.e.* $i$-th element in $z^i$ equals to 1, others equal to 0, $i = 1, 2, ..., K$) representing sub-goal $\hat{g}^i$ in $\mathcal{G}$. We then prompt the same LLM to perform a encoding function $h_{llm}$, which map $s_t$ to latent vector $z_t^{(ref)} \in \mathcal{C}$ by checking whether it belongs to sub-goal $\hat{g}^i \in \mathcal{G}$: $z_t^{(ref)} = h_{llm}(s_t, \mathcal{G})$. We stipulate the output of LLM must be 'yes' or 'no' and then convert it to integer 1 or 0, as this form of answer has shown to be more reliable than the open-ended answer (Du et al. (2023)). By repeatedly asking $K$ times we can finally establish the $K$-dim latent variable $z_t^{(ref)}$ which represents the sub-goal for all $s_t$ in $\mathcal{D}$.

We use these LLM-given latent representations $z_t^{(ref)}$ as supervisory labels for high-level sub-goal encoder training in HIL. Once we obtain these labels, we have no need to interact to LLMs later.

## 4.2 DUAL-ENCODER FOR SUB-GOAL IDENTIFICATION

Naturally, we consider using these LLM-generated labels for sub-goal representations to train a high-level sub-goal encoder $\pi_H(s_t)$ in a supervised manner. Compared to previous unsupervised approaches, this supervised method helps reduce the randomness of output sub-goals by leveraging the guidance provided by the labels. However, it is prone to over-fitting on the training dataset. To address this challenge, inspired by (Ranzato & Szummer (2008); Le et al. (2018b)), we propose a *Dual-Encoder* structure for high-level sub-goal identification. This design integrates both a supervised learning encoder and an unsupervised learning encoder, producing a weighted-average sub-goal representation. The weighted combination allows for flexibility, prioritizing the encoder that performs better for a particular task or dataset, ultimately enhancing robustness and improving generalization.

**Supervised LLM-Label-based Encoder** Considering that the codebook $\mathcal{C}$, representing the sub-goal space $\mathcal{G}$, is discrete and finite, we formulate the supervised sub-goal learning as a multi-class classification problem. To train this supervised learning encoder $\pi_H^{(llm)}$, we define the sub-goal learning objective by maximizing the log-likelihood of the labels generated by the LLMs:

$$\mathcal{L}_H^{(llm)} = \mathbb{E}_{(s_t, z_t^{(ref)}) \sim \mathcal{D}} \ -\log \pi_H^{(llm)}(z_t^{(ref)}|s_t). \tag{1}$$

**Unsupervised VQ Encoder** Given the codebook $\mathcal{C} = \{z^1, z^2, ..., z^K\}$, we apply Vector Quantization (VQ) (Van Den Oord et al. (2017)) to design the unsupervised sub-goal encoder in our SEAL framework. It is a widely used approach that can map the input state $s_t$ to a finite, discrete latent space like $\mathcal{C}$. In VQ, the encoder $\pi_H^{(vq)}$ first predicts a continuous latent vector: $z_t^{(con)} = \pi_H^{(vq)}(s_t)$. This latent vector is then matched to the closest entry in $\mathcal{C}$:

$$z_t^{(vq)} = \text{argmin}_{z^i \in \mathcal{C}} \ \ \|z_t^{(con)} - z^i\|_2^2. \tag{2}$$

The learning objective of $\pi_H^{(vq)}$, named *commitment loss*, encourages the predicted continuous latent vector $z_t^{(con)}$ to cluster to the final output sub-goal representation $z_t^{(vq)}$:

$$\mathcal{L}_H^{(vq)} = \mathbb{E}_{(s_t) \sim \mathcal{D}} \ \ \|\mathbf{sq}(z_t^{(vq)}) - z_t^{(con)}\|_2^2; \tag{3}$$

where $\mathbf{sq}(\cdot)$ denotes stop-gradient operation.

## 4.3 TRANSITION-AUGMENTED LOW-LEVEL POLICY

We compute a weighted-average vector $z_t$ over $z_t^{(llm)}, z_t^{(vq)}$ obtained by dual-encoders to finalize the predicted sub-goal representation:

$$z_t = W_{vq} z_t^{(vq)} + W_{llm} z_t^{(llm)}; \tag{4}$$

where the weights $W_{vq}$ and $W_{llm}$ quantifies *how the predicted sub-goal representations $z_t^{(vq)}$ and $z_t^{(llm)}$ contribute to the task completion success rate.* The weights are updated by validations during the training process. The update details will be demonstrated in Section 4.4.

Given the predicted sub-goal representations $z_t$ for each $s_t$ in the expert dataset, normally the low-level policy agent follows a goal-conditioned behavioral cloning (GC-BC) architecture. It is trained by maximizing the log-likelihood of the actions in the expert dataset, using the sub-goal representations as auxiliary inputs:

$$\mathcal{L}_{GC-BC} = \mathbb{E}_{(s_t, a_t, z_t) \sim \mathcal{D}} \ -\log \pi_L(a_t|s_t, z_t). \tag{5}$$

However, this low-level policy design overlooks the imbalanced distribution and importance of the hierarchical structure captured by high-level sub-goal encoders. Several studies have highlighted that certain states, where transitions between sub-goals occur in long-horizon demonstrations, have a significant impact on the policy agent's performance (Jain & Unhelkar, 2024; Zhai et al., 2022;

Wen et al., 2020). Despite their critical role, these states make up only a small portion of expert demonstrations. Successfully reaching these intermediate states and taking appropriate actions improves sub-goal completion, thereby increasing the overall task success rate. We formally define these states as *intermediate states*:

**Definition 4.3.1. (Intermediate States).** Let $s_t \in \mathcal{S}, 0 \leq t \leq T$ be a state observed when running the HIL agent, $z_t$ is its corresponding latent variable learnt by high-level encoder $\pi_H$ that represents sub-goal. $s_{t+1}$ is the following state. The state $s_t$ is defined as an *intermediate state* only when the sub-goal changes: $z_{t+1} \neq z_t$. Due to the scarcity of these intermediate states, it becomes very challenging to imitate the correct behavior in such states. To address this issue, inspired by the practice of assigning extra rewards to sub-goal transition points in hierarchical RL (Ye et al. (2020); Berner et al. (2019); Zhai et al. (2022); Wen et al. (2020)), we augment the importance of these intermediate states by assigning higher weights to them in the low-level policy training loss:

$$\mathcal{L}_L = \mathbb{E}_{(s_t, a_t, z_t) \sim \mathcal{D}} \quad - e^{\|z_{t+1} - z_t\|_2^2} \log \pi_L(a_t | s_t, z_t); \tag{6}$$

where the term $e^{\|z_{t+1} - z_t\|_2^2}$ measures the L2-distance between the current sub-goal representation $z_t$ and the next sub-goal $z_{t+1}$. Given that $z_t$ is a one-hot vector, we have the term:

$$e^{\|z_{t+1} - z_t\|_2^2} = \begin{cases} e, & \text{if } z_{t+1} \neq z_t \\ 1, & \text{if } z_{t+1} = z_t \end{cases} \tag{7}$$

Thus, this term can serve as an adaptive weight to enhance the imitation of expert behavior at intermediate states. By incorporating this transition-augmented low-level policy design, we emphasize the importance of sub-goal transitions, and in simulations we also observe this design can greatly help agents make transitions across each sub-goal.

## 4.4 TRAINING

We train our SEAL model end-to-end, jointly updating parameters of $\pi_H$ and $\pi_L$ by minimizing the loss function $\mathcal{L} = \beta \mathcal{L}_H + \mathcal{L}_L$, where $\beta$ is a hyper-parameter that controls the weight of high-level sub-goal learning in relation to the overall training process. Additionally, in order to evaluate the reliability of the latent variables predicted by the VQ encoder and LLM-Label-based encoder and determine the weight combination that can better improve task performance, we keep validating the success rates of those two different latent variables in the environment during training. Based on the validation results, we dynamically update the weights $W_{vq}$ and $W_{llm}$ in Eq. 4.

For validation, we simultaneously execute actions conditioned on both the VQ-encoder and the LLM-label-based encoder: $a_t^{(vq)} = \pi_L(s_t, z_t^{(vq)})$ and $a_t^{(llm)} = \pi_L(s_t, z_t^{(llm)})$. We then *run episodes to test the different success rates*, $SR_{vq}$ and $SR_{llm}$, for completing the full task. The updated weights $W_{vq}$ and $W_{llm}$ are then computed as $W_{vq} = SR_{vq}/(SR_{llm} + SR_{vq})$; $W_{llm} = SR_{llm}/(SR_{llm} + SR_{vq})$ respectively. $W_{vq}$, $W_{llm}$ measure the relative task-completion performance of the policy agent under the guidance of $z_t^{(vq)}$ and $z_t^{(llm)}$, respectively. We refer to these weights as *confidences*, indicating the preference for trust between $z_t^{(vq)}$ and $z_t^{(llm)}$.

We also use these weights to finalize the overall training loss of SEAL as a weighted combination of two end-to-end losses under guidance $z_t^{(llm)}$ and $z_t^{(vq)}$. We finalize the overall training loss of SEAL by using a weighted combination of two end-to-end losses, conditioned on $z_t^{(llm)}$ and $z_t^{(vq)}$, with the same weights $W_{vq}, W_{llm}$ determining the contribution of each loss:

$$\mathcal{L}_{vq} = \beta \mathcal{L}_H^{(vq)}(s_t) + \mathcal{L}_L(s_t, z_t^{(vq)}); \mathcal{L}_{llm} = \beta \mathcal{L}_H^{(llm)}(s_t) + \mathcal{L}_L(s_t, z_t^{(llm)}); \mathcal{L}_{SEAL} = W_{vq} \mathcal{L}_{vq} + W_{llm} \mathcal{L}_{llm}. \tag{8}$$

Since the low-level policy agent's actions are conditioned on the latent sub-goal representations, minimizing this weighted-combination loss $\mathcal{L}_{SEAL}$ allows our SEAL to adapt the trainable parameters of the low-level policy based on task-completion performance. This approach helps the agent make better decisions by adjusting to updated latent predictions $z_t = W_{vq} z_t^{(vq)} + W_{llm} z_t^{(llm)}$ during training process. As a result, our SEAL framework can continuously adapt both the high-level sub-goal encoders and the low-level policy agent, leading to more reliable and robust sub-goal representations, as well as improved decision-making for action selection. The complete algorithm for SEAL is illustrated in Algorithm 1.

---

**Algorithm 1** SEmantic-Augmented Imitation Learning (SEAL) via Language Model

---

1: **Input:** Expert Trajectory Dataset $\mathcal{D}$, Natural Language Task Instruction $\mathcal{M}$, Pre-trained LLM $llm$ for sub-goal decomposition and labeling.
2: **Initialize** VQ-encoder $\pi_H^{(vq)}(s_t; \theta_1)$, LLM-Label-based encoder $\pi_H^{(llm)}(s_t; \theta_2)$, Low-level policy agent $\pi_L(s_t, z_t; \theta_3)$, $W_{vq} = W_{llm} = 0.5$ .
3: (**LLM Guiding Sub-goal Learning**)
4: Specify sub-goal space with $\mathcal{M}$: $\mathcal{G} = \{\hat{g}^1, \hat{g}^2, ..., \hat{g}^K\} = f_{llm}(\mathcal{M})$ .
5: Labeling $s_t \in \mathcal{D}$ to latent sub-goal representations $z_t^{(ref)}$: $z_t^{(ref)} = h_{llm}(s_t, \mathcal{G})$ ($z_t^{(ref)} \in \mathcal{C} = \{z^1, z^2, ..., z^K\}$) .
6: (**Training**)
7: **for** Iteration $j$ ($j = 1, 2, ..., J_{max}$) **do**
8:     For $s_t \in \mathcal{D}$, $z_t^{(llm)} \leftarrow \pi_H^{(llm)}(s_t)$,     $z_t^{(vq)} \leftarrow \pi_H^{(vq)}(s_t)$.
9:     Get $\mathcal{L}_H^{(llm)}$ and $\mathcal{L}_H^{(vq)}$ using Eq. 1 and Eq. 3.
10:     $\mathcal{L}_L^{(llm)} \leftarrow \mathcal{L}_L(s_t, a_t, z_t^{(llm)}), \mathcal{L}_L^{(vq)} \leftarrow \mathcal{L}_L(s_t, a_t, z_t^{(vq)})$, using Eq. 6.
11:     $\mathcal{L}_{SEAL} \leftarrow W_{llm}(\mathcal{L}_H^{(llm)} + \mathcal{L}_L^{(llm)}) + W_{vq}(\mathcal{L}_H^{(vq)} + \mathcal{L}_L^{(vq)})$
12:     Update $\theta_1, \theta_2, \theta_3$: $\theta_i \leftarrow \theta_i - \frac{\partial \mathcal{L}_{SEAL}}{\partial \theta_i}$ ($i = 1, 2, 3$)
13:     Validate for: $SR_{(llm)}, SR_{(vq)}$
14:     Update: $W_{vq} = \frac{SR_{vq}}{SR_{llm} + SR_{vq}}, W_{llm} = \frac{SR_{llm}}{SR_{llm} + SR_{vq}}$.
15: **end for**

---

## 5 EXPERIMENTS

In this section, we evaluate the performance of SEAL on two long-horizon compositional tasks: *KeyDoor* and *Grid-World*. We compare SEAL's performance with various baselines, including non-hierarchical, unsupervised, and supervised hierarchical IL methods, in both large and small expert dataset scenarios. Following this, we analyze how SEAL enhances task completion performance.

### 5.1 SIMULATION ENVIRONMENTAL SETUP

**KeyDoor** The MiniGrid Dataset (Chevalier-Boisvert et al., 2018b) is a collection of grid-based environments designed for evaluating reinforcement learning and imitation learning algorithms in tasks requiring navigation, exploration, and planning. Among these environments, we start with *KeyDoor*, an easy-level compositional task that requires the player to move to the key and pick up it to unlock the door. To add complexity, we enlarge the original $3 \times 3$ grid environment to $10 \times 10$ size, and randomly initialize the locations of player, key and door for each episode. To facilitate understanding by LLMs, we convert the environment into a vector-based state, with elements including the coordinates of the player, key, and door, as well as the different statuses of the key (picked or not) and door (locked or not). The maximum time-steps $T$ of one episode is set to 100. We evaluate our SEAL on expert datasets with 30, 100, 150, 200 demonstrations generated by an expert bot.

**Grid-World** The environment is a 10x10 grid world with a single player and multiple objects randomly distributed at various locations. The player's objective is to visit and pick up these objects in a specific order. This task is more challenging than *KeyDoor* due to its longer-range compositional nature, involving more sub-goals. In this work, we set the number of objects in the grid world to range from 3 to 5, to test SEAL's effectiveness in solving longer-range tasks. Similar to *KeyDoor*, the fully observed environment is converted into a vector-based state, with elements representing the coordinates of the player and objects, as well as their statuses (picked or not). The maximum time-steps per episode is set to 100. We evaluate SEAL on expert datasets with 200, 300, and 400 demonstrations generated by an expert bot.

### 5.2 BASELINES

We compare our SEAL with one non-hierarchical learning approach: Behavioral Cloning (BC) (Bain & Sammut, 1995), two unsupervised learning approaches: LISA (Garg et al., 2022) and SDIL

(Zhao et al., 2023), and one LLM-enabled supervised learning approaches Thought Cloning (Hu & Clune, 2024). The detailed settings are listed below:

**Behavioral Cloning (BC)**: The classical imitation learning method with non-hierarchy. We train the policy agent $\pi(a_t|s_t)$ by maximizing the log-likelihood: $\mathcal{L}_{BC} = \mathbb{E}_{(s_t,a_t)\in\mathcal{D}} \quad -\log\pi(a_t|s_t)$.

**LISA**: A HIL approach with unsupervised VQ-based sub-goal learner. We implement the low-level policy agent with only current state $s_t$ as input, rather than a sequence of previous states in original settings, since we assume the task is MDP.

**SDIL**: A HIL approach with an unsupervised sub-goal learner by implementing only the skill discovery component of SDIL, omitting the skill optimality estimation since our expert bot generating demonstrations is optimal. SDIL selects the final sub-goal representation $z_t$ by: $\arg\max_i \quad \frac{1/D(z^i, z'_t)}{\sum_{i=1}^{K} 1/D(z^i, z'_t)}$, where $D$ denotes the Euclidean distance, $z'_t$ is the continuous output vector of $\pi_H$, $z^i \in \mathcal{C}$. To make this selection differentiable for gradient back-propagate, we adopt Gumbel-Softmax (Jang et al., 2016) to replace argmax operation.

**Thought Cloning (TC)**: A HIL approach with supervised sub-goal learner. TC consists of a thought generator $\pi_u(th_t|s_t, th_{t-1})$ (where $th_t$ equals to our $z_t$) and an action generator $\pi_l(a_t|s_t, th_t)$. $\pi_u$ requires supervisory labels training. We apply the LLM-generated sub-goal representations $z_t^{(ref)}$ as labels.

We also present results from two variants of SEAL: **SEAL-L**, which relies entirely on the LLM-label-based high-level sub-goal encoder, and SEAL, which uses the dual-encoder design. SEAL-L is compared with TC to highlight the effectiveness of the low-level transition-augmented design in supervised learning, while SEAL demonstrates the superiority of the dual-encoder approach compared with SEAL-L.

## 5.3 Main Results

We first evaluate the effectiveness of our SEAL approach in two environments: *KeyDoor* and *Grid-World* with 3 objects. These environments are compositional in nature, and the relatively small number of sub-goals simplifies high-level sub-goal encoder learning in HIL settings. To break down the full task instructions into sub-goals, we utilize the latest OpenAI-developed LLM, GPT-4o (Islam & Moushi (2024)). The sub-goal count for *KeyDoor* is $K = 4$, and for *Grid-World* with 3 objects, it is $K = 6$. These identified sub-goals are then used to label states in the expert dataset for supervised high-level encoder training. For a fair comparison, we set the number of sub-goals $K$ in the unsupervised baselines LISA and SDIL to match the LLM-determined sub-goal numbers used in SEAL. Additional details are provided in the Appendix.

| Task | # Traj | BC | LISA | SDIL | TC | SEAL-L | SEAL |
|---|---|---|---|---|---|---|---|
| KeyDoor | 30 | 0.09±0.02 | 0.09±0.02 | 0.23±0.05 | 0.26±0.02 | 0.27±0.06 | **0.30±0.04** |
| | 100 | 0.50±0.06 | 0.53±0.05 | 0.45±0.04 | 0.50±0.03 | 0.52±0.02 | **0.56±0.03** |
| | 150 | 0.67±0.05 | 0.66±0.03 | 0.63±0.05 | 0.69±0.05 | 0.68±0.03 | **0.75±0.04** |
| | 200 | 0.74±0.02 | 0.69±0.04 | 0.70±0.04 | 0.70±0.02 | 0.76±0.04 | **0.82±0.04** |
| GridWorld | 200 | 0.26±0.04 | 0.24±0.03 | 0.43±0.04 | **0.44±0.03** | 0.39±0.04 | 0.29±0.03 |
| | 300 | 0.31±0.04 | 0.44±0.05 | 0.48±0.01 | 0.52±0.07 | **0.65±0.04** | 0.61±0.02 |
| | 400 | 0.48±0.04 | 0.53±0.03 | 0.62±0.04 | 0.62±0.04 | 0.83±0.02 | **0.85±0.02** |

Table 1: **Simulation Results:** Success rates (ranging from 0 to 1) for completing the tasks of *Key-Door* and *Grid-World* (3 Objects), averaged over 5 random seeds. Our SEAL approach outperforms others in most cases. The best-performing method is highlighted in **bold**.

We train the models using randomly sampled expert demonstrations, with 30, 100, 150, and 200 samples for the *KeyDoor* environments, and 200, 300, and 400 samples for *Grid-World* with 3 objects. Since *Grid-World* is a more complex compositional task, we collect additional expert data for this environment. The task success rates from the simulations are presented in Table 1. The results show in most cases, our approach either outperforms or is competitive with the other baselines.

We observe that with fewer training demonstrations, both the HIL baselines and our SEAL outperform non-hierarchical behavior cloning (BC), as they benefit from the additional information learned from hierarchical sub-goal structures. However, as the amount of training data increases, the gap between BC and HIL methods narrows, especially in simpler environments like *KeyDoor*, where BC begins to generalize better due to its exposure to more scenarios. Despite this, SEAL maintains a

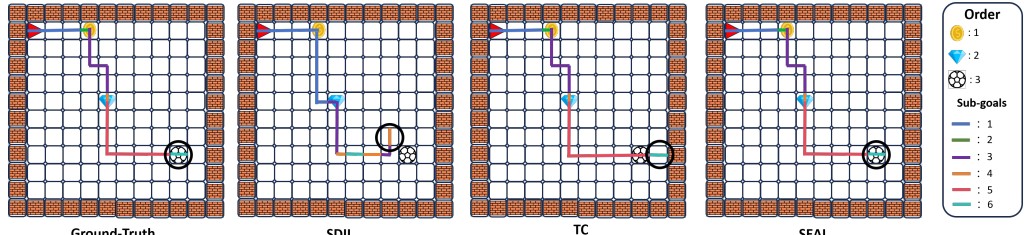

Figure 2: **Visualization:** Sub-goal selection in an example trajectory instance of *Grid-World* with 3 Objects. We color-code each sub-goal and black circle marks the final step of each trajectory. The ground-truth is labeled by human in this case, and SEAL achieve the best sub-goal transitions.

consistent performance advantage do to our Transition-Augmented Low-Level policy agent, which emphasizes imitation the expert actions in challenging sub-goal transition states. As shown in Table 2, SEAL consistently achieves higher sub-goal completion rates, demonstrating its superior ability to select appropriate actions at those critical intermediate states with sub-goal transitions.

| Sub-goals | # Traj | BC | LISA | SDIL | TC | SEAL-L | SEAL |
|---|---|---|---|---|---|---|---|
| Pick up the Key (KeyDoor) | 30 | 0.29±0.05 | 0.22±0.03 | 0.42±0.05 | 0.52±0.04 | 0.55±0.10 | **0.56±0.04** |
| | 100 | 0.78±0.04 | 0.77±0.06 | 0.65±0.02 | 0.67±0.02 | **0.82±0.05** | 0.80±0.02 |
| | 150 | 0.81±0.06 | 0.81±0.03 | 0.80±0.02 | 0.82±0.03 | 0.93±0.02 | **0.93±0.04** |
| | 200 | 0.87±0.01 | 0.88±0.03 | 0.83±0.03 | 0.86±0.01 | 0.97±0.02 | **0.98±0.01** |
| Pick up Object 1 (GridWorld) | 200 | 0.58±0.09 | 0.58±0.05 | 0.79±0.02 | 0.78±0.04 | **0.83±0.04** | 0.67±0.03 |
| | 300 | 0.64±0.06 | 0.71±0.07 | 0.75±0.02 | 0.85±0.03 | **0.90±0.04** | 0.85±0.04 |
| | 400 | 0.75±0.03 | 0.79±0.03 | 0.85±0.03 | 0.87±0.04 | 0.95±0.02 | **0.98±0.01** |
| Pick up Object 2 (GridWorld) | 200 | 0.39±0.06 | 0.36±0.06 | 0.56±0.04 | **0.64±0.05** | 0.61±0.04 | 0.50±0.05 |
| | 300 | 0.44±0.06 | 0.55±0.06 | 0.59±0.03 | 0.52±0.05 | **0.80±0.04** | 0.73±0.03 |
| | 400 | 0.57±0.05 | 0.63±0.04 | 0.70±0.04 | 0.62±0.03 | **0.90±0.02** | 0.89±0.01 |

Table 2: Success rates of sub-goals completion in both *KeyDoor* and *Grid-World*, averaged over 5 random seeds. For the *KeyDoor* environment, the sub-goal is to pick up the key, while for *Grid-World* with 3 objects, the sub-goals are to pick up object 1 and object 2.

We observe that the dual high-level encoder design in our SEAL enhances performance. Compared to relying solely on the LLM-label-based sub-goal encoder, SEAL's dual-encoder design demonstrates higher success rates across both tasks. As shown in Fig. 2, SEAL achieves greater prediction accuracy in the testing environment than other baselines. Unsupervised approaches like SDIL may struggle to accurately capture the ground-truth hierarchical structure, leading the agent to take irregular actions. Meanwhile, Thought Cloning (TC), although typically better at sub-goal prediction accuracy when aided by LLM-given labels, can also fail to specify the sub-goals of crucial states due to overfitting. This can result in invalid actions and ultimately cause task failure.

Additionally, compared to unsupervised HIL baselines, our SEAL removes the burden for tuning the number of sub-goals $K$, by leveraging concrete suggestions from LLMs. Unsupervised methods like LISA, which rely on a pre-defined codebook of sub-goals, can struggle to match the true task hierarchy when choosing inappropriate hyper-parameter $K$. As shown in Fig. 3, both overestimating and underestimating the number of sub-goals can lead to performance degradation. In contrast, our SEAL avoids this issue and outperforms all unsupervised HIL approaches in most cases.

## 5.4 PERFORMANCE ON LONGER COMPOSITIONAL TASKS

| Object Num | # Traj | BC | LISA | SDIL | TC | SEAL-L | SEAL |
|---|---|---|---|---|---|---|---|
| 3 | 300 | 0.31±0.04 | 0.44±0.05 | 0.48±0.01 | 0.52±0.07 | **0.65±0.04** | 0.61±0.02 |
| | 400 | 0.48±0.04 | 0.53±0.03 | 0.62±0.04 | 0.62±0.04 | 0.83±0.02 | **0.85±0.02** |
| 4 | 400 | 0.16±0.03 | 0.13±0.01 | 0.22±0.04 | 0.24±0.05 | 0.26±0.03 | **0.32±0.03** |
| | 500 | 0.39±0.04 | 0.36±0.04 | 0.40±0.01 | 0.39±0.03 | 0.49±0.03 | **0.51±0.03** |
| 5 | 500 | 0.09±0.02 | 0.11±0.04 | 0.11±0.02 | 0.23±0.06 | 0.25±0.05 | **0.42±0.03** |
| | 600 | 0.30±0.03 | 0.47±0.03 | 0.64±0.05 | 0.35±0.03 | 0.65±0.03 | **0.73±0.03** |

Table 3: Success rates on longer-range compositional tasks (*Grid-World*) with 3, 4 and 5 objects.

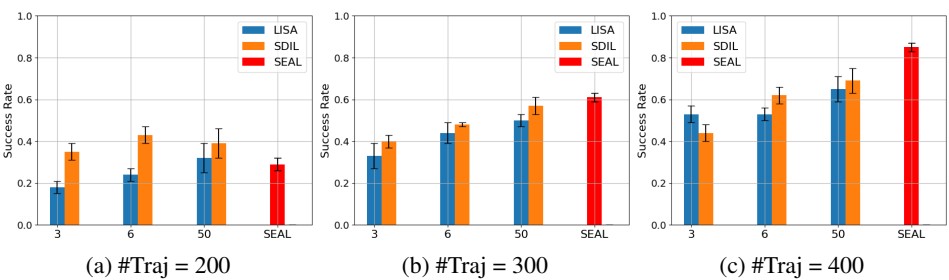

(a) #Traj = 200      (b) #Traj = 300      (c) #Traj = 400

Figure 3: Comparison of success rates among different sub-goal number $K$ selection in unsupervised HIL baselines LISA and SDIL. Experiments set on *Grid-World* with 3 Objects. $x$-axis represents the different settings of $K$.

We further investigate whether SEAL can sustain its effectiveness and superiority on longer-range compositional tasks, which involve more sub-goals. To evaluate this, we test our method on Grid-World with 4 and 5 objects, where the LLMs decompose the task instructions into more sub-goals ($K = 8$ and $K = 10$, respectively). As shown in Table 3, SEAL continues to lead, particularly in cases with smaller expert datasets, demonstrating its adaptability to more complex tasks. In longer-range compositional tasks, managing the increasing complexity of the sub-goal space becomes more challenging, especially for supervised methods like Thought Cloning, as the supervision signal for each sub-goal becomes sparser. SEAL overcomes this by employing a dual-encoder design, which leverages both the flexibility of unsupervised learning to learn sub-goals better. Meanwhile, SEAL focuses more on imitating the few but critical sub-goal transition points in longer-range compositional tasks, avoiding the limitations of signal sparsity faced by other approaches.

## 5.5 ADAPTATION TO TASK VARIATIONS

We also test the adaptability of SEAL to task variations. To do this, we modify the predefined pick-up order in the *Grid-World* environment, which includes three objects: A, B, and C. This generates new tasks for evaluation. We create a dataset comprising 400 expert demonstrations for the task with the order ABC, along with few-shot set of 10 expert demonstrations for other orders such as ACB, BCA, and BAC. We then assess the performance of the trained agent on these alternative orders. As shown in Table 4, our method exhibits slightly higher success rates, indicating that SEAL has better adaptability to task variations. However, it is important to note that this conclusion is limited to specific scenarios. In the grid-world, rearranging the order does not introduce new sub-goals, meaning that the sub-goals learned from the training set remain applicable to these new tasks.

Table 4: Success rates under task variations on *Grid-World* averaged over 5 random seeds.

| Test Env | BC | LISA | SDIL | TC | SEAL-L | SEAL |
|----------|-----|------|------|-----|--------|------|
| ABC | 0.48±0.04 | 0.53±0.03 | 0.62±0.04 | 0.62±0.04 | 0.83±0.02 | **0.85±0.02** |
| ACB | 0.01±0.00 | 0.08±0.02 | 0.11±0.04 | 0.13±0.05 | **0.18±0.07** | 0.14±0.03 |
| BAC | 0.01±0.00 | 0.05±0.01 | 0.06±0.02 | 0.09±0.02 | **0.11±0.03** | 0.08±0.02 |
| BCA | 0.00±0.00 | 0.03±0.01 | 0.08±0.03 | 0.08±0.02 | 0.08±0.01 | **0.09±0.03** |

## 6 CONCLUSION

In this work, we introduce SEAL, a novel HIL framework that leverages LLMs' semantic and world knowledge to define sub-goal spaces and pre-label states as meaningful sub-goal representations without prior task hierarchy knowledge. SEAL outperforms baselines like BC, LISA, SDIL, and TC across various environments, particularly in low-sample and complex longer-range compositional tasks. Our approach achieves higher success and sub-goal completion rates, with assistance of the dual-encoder proving more robust than the pure LLM encoder and the transition-augmented low-level policy. SEAL also adapts well to varying task complexities and latent dimensions. Our current design still observes training instability, and we are interested in making more efficient SEAL approaches while under partially observed states.

## 7 ETHICS STATEMENT

In this work, we develop a new algorithm for hierarchical imitation learning, which builds upon pre-trained large language models (LLMs). The LLMs used in our experiments are based on publicly available GPT-4o API from OpenAI and do not involve any personally identifiable information or sensitive data. The authors are not aware of any additional ethical concerns related to the methodology presented in this research.

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

# A    ADDITIONAL ENVIRONMENT INFORMATION

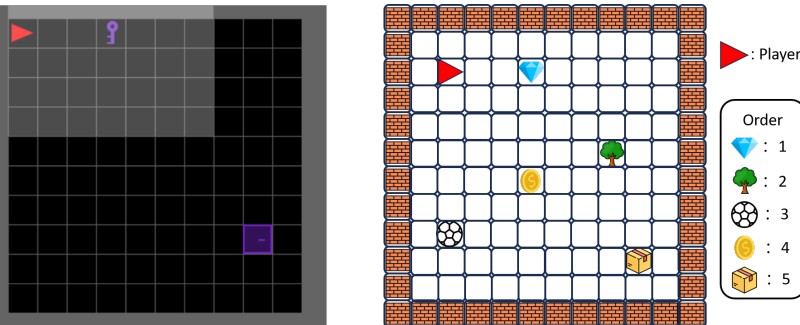

Figure 4: Examples of compositional-task-related environments used in our experiments. **Left:** *KeyDoor*. The player needs to pick up the key and then use it to unlock the door. **Right:** *Grid-World*. The player needs to pick up the different objects in a pre-specified order.

**KeyDoor** The environment is based on the *DoorKey* setting from the MiniGrid Dataset, but with modifications to make the state compatible with LLM input for sub-goal mapping. Instead of using image states, we convert the state into an 8-dimensional vector that captures crucial object information: {x-coordinate of key, y-coordinate of key, x-coordinate of door, y-coordinate of door, x-coordinate of player, y-coordinate of player, key status (picked: 1, not picked: 0), and door status (unlocked: 1, locked: 0) }. Wall obstacles are removed to avoid interference. The action space consists of 6 primitive actions: move up, move down, move right, move left, pick up, and unlock. The key can be picked up only when the player reaches the key's coordinates, and the door can be unlocked only if the player reaches the door's coordinates with the key already picked up. The language task instruction $\mathcal{M}$ is defined as: *"Pick up the key, and then unlock the door."* The episode ends when the door is successfully unlocked or the maximum time steps $T = 100$ are reached.

**Grid-World** The environment is based on the grid world used in (Kipf et al., 2019; Jiang et al., 2022). Similar to *KeyDoor*, the image-based states are converted into a vector format for LLM input, capturing crucial information about objects: x and y coordinates of Object 1, x and y coordinates of Object 2, ..., x and y coordinates of the player, status of Object 1 (picked: 1, not picked: 0), status of Object 2, .... For *Grid-World* with 3, 4, or 5 objects, the state vector has dimensions 11, 14, and 17, respectively. Wall obstacles and irrelevant objects are removed to avoid interference. The action space consists of 5 primitive actions: move up, move down, move right, move left, and pick up. An object can be picked up only when the player reaches its coordinates. The language task instruction $\mathcal{M}$ is defined as: *"Pick up Object 1, then pick up Object 2, then..."* The episode ends when the player picks up all objects in the correct order or after the maximum time step $T = 100$. At the start of each episode, the coordinates of all objects and the player are randomly reset.

**Sub-goal Spaces Identified by LLMs** We use GPT-4o to decompose the language task instructions for both the *KeyDoor* and *Grid-World* environments into their respective sub-goal spaces. In the *KeyDoor* environment, there are $K = 4$ sub-goals: {move to the key, pick up the key, move to the door, unlock the door}. In the *Grid-World* environment, with 3, 4, and 5 objects, the number of sub-goals is $K = 6$, $K = 8$, and $K = 10$, respectively, including: {move to object 1, pick up object 1, move to object 2, pick up object 2, ...}. For both sub-goal spaces, we parameterize each language sub-goal in it by a K-dim one hot vector.

# B    EXAMPLE PROMPTS

In SEAL, we prompt LLMs to generate supervisory labels for training the high-level encoder. Fig. 5 illustrates the detailed prompting process. First, we prompt the LLMs to break down the task instruction into a finite set of sub-goals. Then, for each state, the LLM is prompted $K$ times to determine whether it corresponds to each of the decomposed sub-goals, mapping the states to sub-goal representations. Example prompts for both task decomposition and sub-goal labeling are provided in the following sub-sections.

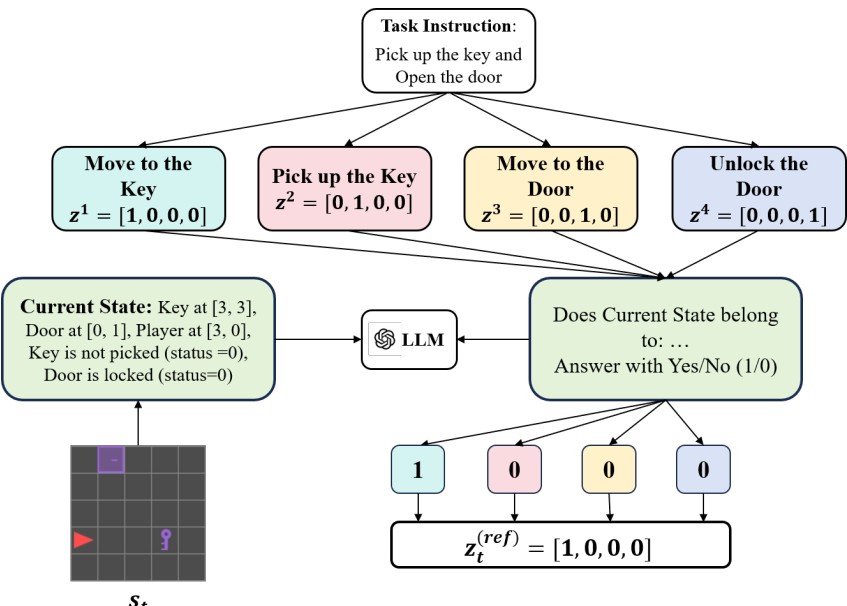

Figure 5: A schematic illustrating how LLMs are prompted to define sub-goal spaces from task instructions and map states to sub-goal representations, serving as supervisory labels for training the high-level sub-goal encoder in SEAL.

## B.1 PROMPTS FOR TASK DECOMPOSITION

An example prompt used in *Grid-World* experiment with 3 objects for task decomposition is listed below. We use GPT-4o to produce the answer.

---

**# Task Description**: Pick up the key, then Pick up the ball, and then pick up the diamond.

**# Environment Details**:

The environment is a 10×10 2D Grid-world.

Objects {Key, Ball, Diamond, Player} (Key, Ball, and Diamond's coordinates are fixed, The player can move)

Observation Space: {o1: The coordinate of the key
o2: The coordinate of the ball
o3: The coordinate of the diamond
o4: The coordinate of player itself
o5: The status of key (picked/not)
o6: The status of ball (picked/not)
o7: The status of diamond (picked/not)}

Action Space: {move up/right/left/down, pick up}

**# Role Instruction:**
You should give an abstract plan to solve the given task step-by-step. For each step in plan, you need to extract relevant features in observation space. You should answer in format:

{Step 1:..., Relevant Features:{o1,....} , Step 2:..., Relevant Features:{o1,....} ...... Step N: ..., Relevant Features:{o1,....} }

---

The GPT-4o answer with:

Here is the abstract plan to solve the task step-by-step, with relevant features in the observation space for each step:

**Step 1: Move the player to the key's coordinate.**
Relevant Features: o1: The coordinate of the key. o4: The coordinate of the player.

**Step 2: Pick up the key.**
Relevant Features: o1: The coordinate of the key. o4: The coordinate of the player. o5: Status of the key (picked/not).

**Step 3: Move the player to the ball's coordinate.**
Relevant Features: o2: The coordinate of the ball. o4: The coordinate of the player. o5: Status of the key (picked/not).

**Step 4: Pick up the ball.**
Relevant Features: o2: The coordinate of the ball. o4: The coordinate of the player. o6: Status of the ball (picked/not).

**Step 5: Move the player to the diamond's coordinate.**
Relevant Features: o3: The coordinate of the diamond. o4: The coordinate of the player. o5: Status of the key (picked/not). o6: Status of the ball (picked/not).

**Step 6: Pick up the diamond.**
Relevant Features: o3: The coordinate of the diamond. o4: The coordinate of the player. o7: Status of the diamond (picked/not).

This step-by-step plan ensures that the player collects the objects in the specific required order: key, ball, and then diamond.

## B.2 PROMPTS FOR MAPPING STATES TO SUB-GOAL REPRESENTATIONS

An example prompt used in *KeyDoor* experiment for mapping states in expert demonstration to sub-goal representations is listed below. We use GPT-4o to produce the answer.

**# Environment:**
A 2D-grid World with a key and and a door in it. The grid world's size is 10 * 10. The coordinate in grid is written as [x, y] (x=0,1,2,3,4, y=0,1,2,3,4) Key at the coordinate: [3, 3]. Door at the coordinate: [0, 1]. Key is not picked up. (key state = 0) Door is locked. (door state = 0)
(Hint: You can only pick up the key when you at the key location, and can only unlock the door when you are at the door location and have already picked up the key).

**# Current State:**
The player is currently at [3, 0]. The key state = 0. The door state = 0. The next coordinate of player is at [3,3].

**# Sub-goal Judgement:**
You need to judge which stage the player in:
1. The player should move to the Key.
2. The player should pick up the Key.
3. The player should move to the Door.
4. The player should unlock the Door.
Please check whether the current state in the above stages in turn. For each judgement please answer with 1(Yes) or 0 (No).

You should finally answer with a 4-dimension vector format: [1/0, 1/0, 1/0, 1/0]

The GPT-4o response with:

> The current state of the player is at [3, 0] and the next coordinate is [3, 3], which is the location of the key.
>
> Here's the judgment for each state:
>
> The player should move to the Key: 1 (The player is moving towards the key)
> The player should pick up the Key: 0 (The player is not yet at the key location)
> The player should move to the Door: 0 (The player has not picked up the key yet)
> The player should unlock the Door: 0 (The player needs to pick up the key first)
>
> So, the 4-dimension vector is: **[1, 0, 0, 0]**

## C  MODEL IMPLEMENTATION DETAILS

We outline the model implementation details for all four baselines and SEAL in the *KeyDoor* and *Grid-World* environments. For non-hierarchical BC baselines, we use a two-layer Multi-layer Perceptron (MLP) as the trainable policy agent $\pi(a_t|s_t)$. In HIL approaches like LISA, SDIL, and TC, this same two-layer MLP is used for both the high-level sub-goal encoder $\pi_H(s_t)$ and the low-level policy agent $\pi_L(a_t|s_t, z_t)$. We use Adam (Kingma, 2014) as the optimizer for all models, with learning rates initialized at 5e-5 for *KeyDoor* and 5e-6 for *Grid-World*. To ensure fair comparison, we maintain consistent hyper-parameters across all simulations, including the high-level encoder loss weight $\beta$, the hidden dimensions of the MLPs, and the number of sub-goals $K$ for both HIL baselines and SEAL. Detailed implementations are presented in the following Table 5:

| Methods | Loss | Hidden dim of MLPs | $\beta$ |
|---|---|---|---|
| BC | $\mathcal{L}_{BC} = \mathbb{E}_{(s_t,a_t)\in\mathcal{D}} - \log\pi(a_t|s_t)$ | [128, 128] | / |
| LISA | $\mathcal{L}_{LISA} = \beta\mathcal{L}_H^{(vq)}(s_t) + \mathcal{L}_L(s_t, z_t^{(vq)})$ | [128, 128] | 0.4 |
| SDIL | $\mathcal{L}_{SDIL} = \mathbb{E}_{(s_t,a_t)\in\mathcal{D}}\mathbb{E}_{z_t\in\pi_H(z_t|s_t)} - \log\pi_L(a_t|z_t, s_t)$ | [128, 128] | / |
| TC | $\mathcal{L}_{TC} = \mathbb{E}_{(s_t,a_t,z_t)\in\mathcal{D}} - \log(\beta\pi_H(z_t|z_{t-1}, s_t) + \pi_L(a_t|s_t, z_t))$ | [128, 128] | 0.4 |
| SEAL | $\mathcal{L}_{SEAL} = W_{llm}(\mathcal{L}_H^{(llm)} + \mathcal{L}_L^{(llm)}) + W_{vq}(\mathcal{L}_H^{(vq)} + \mathcal{L}_L^{(vq)})$ | [128, 128] | 0.4 |

Table 5: Hyperparameters settings of Model Implementations.

