# OpenReview forum: "SEAL: SEmantic-Augmented Imitation Learning via Language Model"
_ICLR.cc/2025/Conference — Submitted to ICLR 2025_

### Official Review · Reviewer_Db38 · 2024-11-02

**Soundness:** 4
**Presentation:** 3
**Contribution:** 3
**Rating:** 6
**Confidence:** 4

**Summary:**

SEAL  introduces a novel framework leveraging LLMs to enhance Hierarchical Imitation Learning (HIL) by addressing long-horizon decision-making tasks.

SEAL utilizes LLMs to generate semantically meaningful sub-goal representations, thus eliminating the need for extensive expert demonstrations and detailed supervisory labels. It employs a dual-encoder structure combining supervised LLM-guided sub-goal learning with unsupervised Vector Quantization for robust sub-goal representations.

Additionally, SEAL features a transition-augmented low-level planner to improve adaptation to sub-goal transitions. Experimental results indicate that SEAL outperforms existing HIL methods, especially in scenarios with limited expert data and complex tasks.

**Strengths:**

SEAL effectively leverages LLMs to autonomously define and label sub-goals, enabling the framework to operate without detailed task hierarchy knowledge. I believe this is a crucial step to eliminate the needs to extensively labeling.

The dual-encoder system, combining LLM-based supervision with Vector Quantization, strengthens sub-goal representation and improves model robustness.

SEAL desires several easy but non-trivial tasks to prove it effectiveness. The  transition-augmented low-level planner adapts well to intermediate task transitions, enhancing performance on complex and long-horizon tasks.

**Weaknesses:**

SEAL heavily relies on the semantic abilities of LLM's quality to generate meaningful sub-goals and guide the hierarchy.

Need more complicated envs. An env with continuous space and action could prove more.

**Questions:**

See weaknesses.

---

### Official Review · Reviewer_6UAj · 2024-11-04

**Soundness:** 2
**Presentation:** 2
**Contribution:** 2
**Rating:** 3
**Confidence:** 4

**Summary:**

This paper tries to leverage LLM to improve sub-goal learning from unsegmented demonstrations for hierarchical imitation learning. The proposed method SEAL combines supervised subgoal learning using LLM labels and unsupervised subgoal learning using vector quantization. The method is tested on two discrete environments KeyDoor and Grid-World with improved results over baselines.

**Strengths:**

The paper works on an important problem of learning semantically meaningful subgoals for demonstrations without dense expert labels. The attempt to leverage LLM knowledge about discrete task structure is well-positioned, and authors took the effort to compare SEAL to many baselines.

**Weaknesses:**

The biggest concern I have for the paper is that the algorithm 1 looks very complex yet the experiments are only two simple discrete grid-world domains. I am not fully convinced why such simple domains require such sophistication in loss term designs while there exist simpler algorithms for more complex domains [1, 2]. Perhaps you should consider citing them, explaining the differences in related works or using them as potential baselines, which might be more relevant than your current baselines. While I understand you have motivated why you need each loss term through high-level description, you should show concrete examples or ablation results to support the claims more clearly.

[1] Learning Rational Subgoals from Demonstrations and Instructions
[2] Grounding Language Plans in Demonstrations Through Counterfactual Perturbations

**Questions:**

1. Why is LLM necessary? What's the benefit of using LLM over abstract task descriptions provided by humans [1]? Do you have quantitative or qualitative results supporting your claim?
2. Assuming LLM is necessary, why is each component of SEAL necessary to learn semantically meaningful subgoals? Can you not use simpler learning objectives [2] to achieve the same goal?
3. What are concrete examples to highlight the drawbacks of using supervised loss or unsupervised loss alone? How to ensure that during training supervised learned goals and unsupervised learned goals are aligned so the weighted average make sense?
4. How do you use LLM to map states to subgoals? While I understand LLM can propose a discrete set of subgoals, but how do you prompt LLM to map states to one of these subgoals? Do you have concrete examples to clarify the mechanism?
5. How do you know where the intermediate states are so you can apply transition adaptive weights?
6. What are the motivations for using each of the current baselines? What claim does comparison with each of the baselines support?
7. Could you include more complex experiments or domains to show the benefits of your approach more convincingly?

---

### Official Review · Reviewer_S24f · 2024-11-04

**Soundness:** 4
**Presentation:** 3
**Contribution:** 3
**Rating:** 5
**Confidence:** 5

**Summary:**

The SEAL framework (Semantic-Augmented Imitation Learning via Language Model) addresses the challenges of Hierarchical Imitation Learning (HIL) in long-horizon decision-making tasks by leveraging the semantic and reasoning capabilities of Large Language Models (LLMs). SEAL utilizes LLMs to generate meaningful sub-goal representations from language instructions, automatically defining sub-goal spaces without prior task hierarchy knowledge. A dual-encoder structure combines supervised LLM-guided learning with unsupervised vector quantization, improving sub-goal representation reliability. Additionally, SEAL’s transition-augmented low-level planner enhances policy adaptation to sub-goal transitions. SEAL demonstrated improved performance over existing HIL and LLM-based planning methods in tasks requiring complex, long-horizon planning, especially with limited expert datasets.

**Strengths:**

The strength of this paper lies in its innovative use of Large Language Models (LLMs) for sub-goal generation and labeling within imitation learning, moving beyond the typical application of LLMs as plan generators. By using LLMs to derive semantically meaningful sub-goal commands and annotate demonstration trajectories, the authors introduce a unique way of harnessing LLMs for hierarchical task decomposition. This labeling mechanism directly supports policy training, effectively guiding the learning process in complex, long-horizon tasks without relying on predefined task hierarchies or extensive manual labeling. This approach highlights the potential of LLMs to provide structured supervision in imitation learning.

**Weaknesses:**

The paper has several limitations. First, it claims using Vector Quantization (VQ) as part of a key contribution, but there’s no abaltion studies to show if VQ really helps. It would be more convincing if there was an experiment testing VQ against an ablated model that does not have quantization to see if it makes a difference.

Second, the evaluations were done in simple environments with only a few objects (n = 3), which raises concerns. In larger or more complex environments, VQ might not scale well. Also, using LLMs to classify states based on natural language descriptions could cause problems if there are many similar-looking objects (e.g., think about a Sokoban environment where there are many boxes of the same shape/color on the ground) or if the environment has continuous states and actions, like in robotics. Using language as the intermediate representation might not handle these situations "accurately".

Based on the Appendix, the prompts used for SEAL are very specific to the environments tested, which makes it unclear if they would work well in other, different environments without adjustments. Or, how should a user decide on what to include in the prompts?

Finally, if data efficiency is a main advantage of SEAL, it should be compared to other methods for data-efficient RL/IL, like Glide (https://arxiv.org/abs/2403.17124 which also uses LLMs to decompose tasks) and EfficientZero (https://arxiv.org/abs/2111.00210 which focuses on data-efficient model-based RL). Comparing SEAL to these would make it clearer how it performs against other approaches focused on efficient learning.

**Questions:**

How did you pick the codebook size for the VQ module? How does that affect the model performance?

---

### Official Review · Reviewer_ZZWW · 2024-11-05

**Soundness:** 3
**Presentation:** 3
**Contribution:** 2
**Rating:** 3
**Confidence:** 4

**Summary:**

The authors present a hierarchical imitation learning method that tries to learn a latent subgoal representation.
The subgoal representation is learned from
1. Generating language subgoals from instructions using LLM prompting.
2. Labeling the state from the expert demonstration with the generated sub-goals using LLM prompting.
3. The labeled states is used to train a supervised subgoal encoder, while the unlabeled state gets used to train a VQ encoder using the subgoal codebook.

The proposed method shows improvement on 2D grid world environment on task success rate.

**Strengths:**

- Authors study an important problem of representing subgoals from task instructions.
- Automatically does state-sub goal matching from unlabeled trajectory.
- The author's experiments with both supervised subgoal encoder and unsupervised VQ encoder is an interest to the readers, but it is up to debate whether having both is beneficial.
- Experiments are comprehensive.

**Weaknesses:**

Author's motivation for the work is quite weak. The authors claim that they want to ask three questions regarding whether pre-trained LLMs can:
1. Serve as a prior to define a task’s hierarchical structure,
2. Establish a sub-goal library autonomously, and
3. Closely guide both high-level sub-goal learning and low-level agent behavior.

There are many existing works (e.g., Voyager, CoT, ToT, LLM-P, etc.) that demonstrate LLMs excel at points 1 and 2, so the main problem is how the authors can make a meaningful contribution to point 3. However, the contributions regarding point 3 are quite limited. The explanation is as follows:

Method:
- The method can be understood as learning an encoder to output a latent representation of a subgoal (which is part of a skill library), a task that existing LLMs already excel at.
- The method appears highly dependent on the quality of LLM-generated state-subgoal matching, which might be manageable in a simple environment and task like those in the paper but is unlikely to transfer well to more complex environments involving multimodal input. The difficulty of such state-subgoal matching has been explored in multiple works (e.g., 1, 2 and more) in robotics.
- The only way the learning of the low-level agent is impacted is by assigning higher weight to states with subgoal transitions. This is a minor contribution.

Experiments:
- The environment and task are quite simple, as the language instructions can be easily decomposed into a limited set of subgoals.
- Although the environment is 2D, it is represented as text, which severely limits the applicability of this method, as the application will be constrained by how the environment can be represented as text.

1. LOTUS: Continual Imitation Learning for Robot Manipulation Through Unsupervised Skill Discovery
2. Skill-based Model-based Reinforcement Learning

**Questions:**

- How does the learned subgoal representation z_t compare to a representation generated by a LLM? What is the benefit of having a dual encoder compared to simply using open-source LLMs as Llama to represent the subgoals?
- How does this method compare to the rich literature of methods that work on automatic task decomposition and skill library generation (e.g. Voyager)?
- How does the authors propose to do subgoal-state matching when the environment gets more complex?
- How does the authors propose to do task decomposition when the environment is partially observable? Some initially-generated subgoals may not be valid and be changed on the go if the environment can not be fully observed.
- How is the learned subgoal representation actually benefit the low level policy other than assigning higher weights to transition states? Why is there no ablation on this?

Check the weaknesses section for more comments.

---

### Meta-Review · Area_Chair_rTL2 · 2024-12-21

**Metareview:**

The authors present a hierarchical imitation learning method that tries to learn a latent subgoal representation. The subgoal representation is learned from by (1) generating language subgoals from instructions using LLM prompting. (2) labeling the state from the expert demonstration with the generated sub-goals using LLM prompting. (3) the labeled states is used to train a supervised subgoal encoder, while the unlabeled state gets used to train a VQ encoder using the subgoal codebook. (4) The overall proposed method shows improvement on 2D grid world environment on task success rate.

While the paper studies an important problem the overall motivation of the work as well as the need of using LLM is not well explained.

**Additional Comments On Reviewer Discussion:**

The authors did not respond to reviewer concerns.

---

### Decision · Program_Chairs · 2025-01-22

Reject